# Interventions by Caregivers to Promote Motor Development in Young Children, the Caregivers’ Attitudes and Benefits Hereof: A Scoping Review

**DOI:** 10.3390/ijerph191811543

**Published:** 2022-09-14

**Authors:** Marlene Rosager Lund Pedersen, Anne Faber Hansen

**Affiliations:** 1Department of Sports Science and Clinical Biomechanics, Faculty of Health Sciences, University of Southern Denmark, Campusvej 55, 5230 Odense M, Denmark; 2Department of Research and Analysis, University Library of Southern Denmark, 5230 Odense M, Denmark

**Keywords:** infants, young children, toddlers, motor development, motor skills, scoping review, interventions, caregivers, parents

## Abstract

In the first year of life, the child’s caregivers, including parents and daycare staff, play an essential role, as they are responsible for implementing daily activities to promote the motor development of young children. However, what does the research show about interventions to promote the motor development of 0–36-month-olds carried out by the child’s caregivers, and what are the caregivers’ experiences and attitudes hereof? This scoping review aims to provide an overview of the published studies to derive an overall interpretation. A systematic search was conducted in five scientific databases, resulting in 10,219 articles, of which 9 met the inclusion criteria. The results indicate that providing early intervention to 0–36-month-old children, in which the caregivers carry out the activities, promotes the young child’s motor development. Furthermore, the interventions increase the caregivers’ interest and motivation to promote the young child’s motor development, which is essential in maintaining the behaviour after the end of the interventions. Supervision and guidance provided for the child’s caregivers concerning knowledge and skills about age-appropriate behaviours and facilitation of their child’s motor development increases the caregivers’ self-confidence, interest, and motivation.

## 1. Introduction

The literature shows that improved motor competence leads to more physical activity [1], better respiratory fitness [2], improved cognitive development [3], social development [4] and better language acquisition [5]. Children with better motor skills are also more likely to choose physical leisure activities later in life [6,7,8]. Conversely, children with motor difficulties are more likely to have lower self-esteem [9] and higher levels of anxiety [10].

In Denmark, a study including 16,686 children born in 2017 shows that 10.1% of these 16,686 children aged 8–10 months have received a motor development ‘observation’ [11]. An ‘observation’ indicates that the motor development does not meet the expected development standard in line with the child’s age [11]. Furthermore, studies in Denmark show that motor difficulties in the early years of children’s growth lead to an increased risk of motor difficulties at the start of school [12,13]. Thus, there are solid arguments for promoting young children’s motor development, as it has many positive effects both in the short and long term for the child.

Studies have shown that in the early years of a child’s life, caregivers play an essential role in improving motor development in young children by being role models and by providing opportunities for movement, encouragement and support [14,15,16,17,18], as well as by reducing inappropriate behaviour, such as understimulation or passivity [18,19]. For example, one study shows that infants spend too much time on their backs in car seats, strollers and similar restrictive devices [20]. Activities and exercises for the young child require adult involvement, as they must implement daily activities to promote the children’s motor development. For example, the caregiver may place a young child on his/her stomach to promote motor development [18,21,22], as this strengthens the infant’s muscles for motor milestones, such as controlling the head, reaching, crawling and pulling him/herself up [18], and tummy time is associated with motor development [23]. WHO also recommends that young children participate in daily activities that promote the child’s motor development, such as tummy time, rolling and crawling, in open and safe play areas where infants can participate in free movements with appropriate toys, such as rattles and balls [24]. In addition, it is essential to limit the time spent in restrictive devices, such as car seats or reclining chairs [24].

As shown above, the daily caregivers play an essential role in supporting and promoting the young child’s motor development. The young child’s primary caregivers are first and foremost the parents, followed by the educational staff in the daycare centres where the child spends many hours during its first years of life.

Interventions aimed at the caregivers to promote the young child’s motor development are interesting in terms of observing the caregivers’ experience and attitudes towards the interventions.

It is both essential to see the interventions’ impact on the young child’s motor development and the significance of this on the caregivers’ actions and attitudes. This knowledge may provide an interpretation of the field where caregivers try to promote young children’s motor development. To our knowledge, no systematic synthesis of studies involving 0–36-month-old children has been done previously. Previous systematic reviews have included studies of older children aged three to six years [14,15,25] or premature infants [26].

## 2. Aim

The primary aim of this review is to bring together the available evidence about interventions that have been implemented to improve 0–36-month-olds’ motor development, which has been performed by the child’s primary caregivers, who are defined in this article as the child’s parents and the educational staff in the childcare setting. The secondary aim is to determine the caregivers, experiences, benefits and attitudes toward the interventions.

The primary outcome of the study is to describe the impact of the interventions on the young child’s motor development and, secondly, to shed light on the caregivers’ attitudes towards the interventions.

## 3. Methods

### 3.1. Identification of Article

The purpose of this scoping review is to identify the research that exists in this area in order to provide a descriptive summary [27]. We used a review method described by Peters et al. [28]. Initially, we conducted a search of the core concepts of the research question in several scientific databases, and the databases that showed the most relevant search results were selected for the final search. The search strategy was developed in collaboration between the researcher and a research librarian.

The literature search was conducted in ERIC, APA PsycInfo, Scopus, SPORTDiscus and Web of Science. Studies published between 1 January 2000 and 6 April 2022 with full text in either Danish, Swedish, Norwegian, English or German were included. Duplicates were removed before starting the review process. The review process followed the inclusion and exclusion criteria (see Table 1).

### 3.2. Search Strategy

The search strategy used the Boolean operators “AND” and “OR”. Proximity operators were used where appropriate, e.g., (N6, adj6, or W/6) to allow up to two words and indifferent ordering of keywords. Truncations (*) were used to ensure both singular and plural forms of keywords. The search terms were a mixture of free text keywords and subject keywords (marked with “DE” in SPORTDiscus and ERIC, and with “/”, or “exp.../” in APA Psycinfo). The search was supplemented with chain searches both backwards (in reference lists) and forwards (Citation Index in Scopus). The search strategies in the different databases were as similar as possible to the extent permitted by subject word and syntax differences. The entire search strategy can be seen in Table 2.

### 3.3. Screening and Selection of Articles

Articles that met the study’s inclusion and exclusion criteria were included. The articles should include some kind of measurement of the intervention’s impact on the children’s motor development, which is the primary outcome measure in the article. The included articles were imported into Covidence and screened for title and abstract. The articles that were not screened out were then screened at the full-text level, and the excluded studies were done for a reason. In the screening process, two people screened the articles independently. Before starting the actual screening process, both authors underwent validation by screening the first 50 articles together, after which both authors screened the articles independently. In the case of disagreement, the authors reviewed the articles together and reached consensus.

### 3.4. Extraction and Interpretation of Data

The final and systematic search strategy, which evaluated interventions targeting 0–36-month-olds’ motor development, performed by the caregivers of the young child and their benefits of and the attitudes towards the intervention, was followed by a mapping of the characteristics of the identified studies, including study type, study population, age, intervention, duration, variables, primary outcome measure (the child’s motor development) and secondary outcome measure (the carers’ attitudes to and their benefits of the intervention). Based on the mapping of characteristics in the different articles, a summary was made, followed by an interpretation of what is shown in the literature as a whole.

As this was a scoping review, the aim of which is to identify the research that exists in the field, no quality assessment of the articles was carried out [27].

## 4. Results

### 4.1. Number of Included Studies

A total of 10,219 articles were identified in the five databases. After removing duplicates in Endnote and importing the search results into Covidence, where another duplicate removal was performed, 5832 articles were screened for title and abstract. Of these, 5779 articles were deemed irrelevant in relation to the described inclusion and exclusion criteria (illustrated in Table 1).

A total of 53 articles were reviewed in full text, after which 44 studies were excluded for one of the following reasons: wrong population, outcome, intervention, country, study design, and language according to the inclusion and exclusion criteria of this study, see Table 1. Six articles were included for analysis. A chain search found three additional articles, and all nine articles were included in the analysis [29,30,31,32,33,34,35,36,37]. See the screening process in Figure 1.

### 4.2. Description of the Included Studies

To extract data from the articles, a table was created, encompassing the categories: first author, year of publication, country, study type, study population, age, intervention type, duration, variables and outcomes. All nine included articles met the study’s inclusion and exclusion criteria and reported on interventions performed by a daily caregiver to promote the motor development on 0–36-month-old young children. Six studies were carried out in the United States, two in Australia and the last in Switzerland. Five of the studies were randomised controlled trials, three were non-randomised controlled trials and the last was an observational study with intervention. Across the articles, there were a total of 960 young children aged 0–36 months. The study-populations ranged from 7 to 456 young children. Six of the studies concerned an intervention for young children under 1 year of age, carried out by a caregiver of the child. Three studies concerned young children above one year of age, where a caregiver carried out the intervention. All nine studies reported some form of motor development of the young child, and in six of the studies the caregivers’ attitudes and benefits of the intervention were also reported. In all studies, the motor development outcome was measured immediately after the intervention ended, and three of the studies also measured the long-term effects on the young child’s motor development using follow-up measurements [30,31,36]. For detailed information, see Table 3.

### 4.3. Summary of Results

#### 4.3.1. Impact of Interventions on the Young Child’s Motor Development

Overall, as seen in Table 3, column 7, the studies show that the interventions promote the young child’s motor development, compared with the control group, both immediately after the intervention and in the later follow-up measurements. A general element across the included studies was that the caregivers received some form of knowledge, guidance and practical experience with activities they carried out with the young child. In three of the studies, the interventions beside supervision of the caregivers also included the use of toys (e.g., a rattle or a grasp ball) [31,35,37]. However, there were differences in how much supervision the caregivers received, the dosage of how much they must train with their child and the length of the interventions. The interventions varied from 1 week [37] to 36 months [34]. The interventions reported impact on the young child’s motor development, as well as on the behaviour of the caregivers who must carry out the training.

In the studies involving young children under one year, the focus was on activities related to prone positioning/tummy time, head control and strengthening of the neck, back and body muscles, performed by the child’s caregiver [29,30,31,32,36,37].

In the studies including young children older than one year, there was focus on giving caregivers knowledge and promoting different activities with their child [33,34,35]. One study used instructional videos to engage and give feedback to the caregivers [35], another study used a training program that the caregivers should perform for 10 min daily [33] and in a third study the caregivers had professional staff visit once a month where new activities were performed [34]. One of the included studies investigated two interventions on the young child’s motor development, one guided by a study director and the other guided by the young child’s daily caregivers. The study showed that when the caregivers performed the intervention, it improved the young child’s motor development and reduced the young child’s negative vocalization during the activities, such as whining, crying and showing dissatisfaction. This was not the case in the intervention carried out by the study director [32]. This points toward the idea that an intervention led by caregivers affects the young child’s motor development more, and there is more satisfaction during the activities, compared to the intervention led by a study director. Thus, the social value and attachment to the person seems important [32].

#### 4.3.2. Caregivers’ Attitudes and Benefits of the Interventions

As shown in Table 3 column 8, six of the studies held information about caregivers’ attitudes, benefits and experiences with the intervention [30,32,33,34,35,37]. In general, the caregivers were satisfied with the interventions and found them meaningful and relevant. In four of the studies, it is reported that the caregivers were positive and motivated concerning the intervention [30,33,36,37]. Furthermore, the interventions lead to a greater interest in play, activities, and the use of toys with the young child, even after the intervention [30,35]. In addition, one study also showed a stress reduction among the caregivers [35]. In two of the studies, the caregivers reported that the activities and the efforts were relevant and valuable [32,36].

A general overview of the results is presented in Table 4.

## 5. Discussion

This scoping review presents an overview of the impact of early interventions on improving motor development in 0–36-month-old children where the caregivers performed the interventions, as well as the caregiver’s experience, benefits and attitudes towards the interventions. The studies show that interventions performed by a caregiver improved motor development immediately after the intervention and during follow-up. The [30,32,33,34,35,37] studies that reported the caregiver’s experience, benefits and attitudes towards the interventions generally showed positive attitudes towards the interventions and increased interest and motivation in children’s motor development.

This review’s finding—that early interventions impact motor development in 0–36-month-old children—can also be seen in studies involving older children. Three reviews involving children of preschool age (3–5 years) show that interventions for this age group promote children’s motor development [14,15,25]. Moreover, studies on premature infants find that early intervention positively affects children’s motor development [26]. Thus, this review’s findings align with earlier studies on other target groups.

An earlier study describes that facilitators for conducting activities and tummy time with their children depend on the caregivers’ motivation, self-efficacy, and knowledge [38]. On the other hand, a lack of caregivers’ motivation, self-efficacy and knowledge may be barriers [38]. Previous studies confirm that knowledge and information are essential for caregivers to know the current recommendations [14,18,26]. For example, a study by Zachry and Kitzmann [18] shows that many caregivers are unaware of the importance of placing the young child in a prone position/tummy time or the complications that can arise if this is not done [18]. This study’s findings align with previous studies that recommend improving interventions, giving knowledge and training caregivers to activate young children [30,39]. Thus, one of the main components of early interventions is that they target the young child’s caregivers to act and acquire skills to promote the child’s motor development [18,40]. Therefore, many studies highlight the importance of professional staff training, informing the child’s caregivers about the importance of activities/play and avoiding or reducing inappropriate behavioural patterns, such as pacification [14,18,19,26,41]. Other studies show that when caregivers know how to accomplish correct behaviour, such as putting their young child in a prone position, they feel competence or self-efficacy [37]. The more the caregivers are associated with positive emotions, the more the caregivers practice and ultimately have a stronger caregiver–child relationship [42]. Bandura described how success in learning a skill, repeated over multiple successes, leads to higher confidence and a sense of competence not merely with the involved skill but also more generally in life [43]. Less success decreases self-efficacy, mainly if failures are experienced before developing a general sense of efficacy. Early parenthood is thus a salient stage for supporting a sense of efficacy [43]. Providing caregivers with knowledge and activities they can use to promote the young child’s motor development boosts their sense of caregiver efficacy.

Three of the included studies in this review used toys in the intervention [31,35,37] besides knowledge, guidance and practice. Earlier studies showed that using toys in an intervention can affect the caregivers’ willingness and curiosity to do the activities [44]. Other studies found that if appropriate tools and information were provided, even the more socially vulnerable parents would act on the knowledge [44,45,46]. Regarding collaboration with the persons who give the intervention (e.g., childcare providers), toys may have a good impact on both the caregivers’ collaboration, willingness, and the motivation to learn, which may promote the young child’s motor development [47]. In addition to knowledge, lack of finances is also a barrier to doing activities with young children [48]. Children from lower socioeconomic groups are more likely to be developmentally delayed [49], so toys may be a recommendation in interventions, especially for these groups.

This review cannot conclude recommendations for the length of a given intervention. However, based on this study’s findings, the following elements may be advantageously included in interventions: supervision for the child’s caregivers to acquire knowledge and skills about age-appropriate behaviours in relation to their young child’s motor development, as well as to practice age-appropriate training to build up skills and concretely acquire ideas for implementation. In addition, knowledge about age-appropriate toys is essential in the interventions. The child’s daily caregivers’ interaction is essential, as a trusted person performing training impacts the young child’s motor development [32]. A recommendation would, therefore, be that when designing interventions for 0–36-year-olds, it must be performed by the child’s caregivers.

### 5.1. Methodological Considerations Relating to the Articles Included in this Review

In general, the results of this review should be interpreted with some caution. The results are based on relatively few respondents, and due to the “natural experiments” study designs, the studies do not report which other activities caregivers engage in with their children. This may influence the conclusions made. Another weakness of the review is the credibility of our interpretation of the articles, as it is based on only nine articles. The fact that the included studies have an experimental design in which the intervention groups are compared with a control group helps to strengthen credibility. However, the use of RCT studies only, would have strengthened the conclusions.

### 5.2. Methodological Considerations of This Review

The strengths of this scoping review are that the literature search method was comprehensive, systematic and conducted in five databases in collaboration with a trained research librarian. The search has provided insight into the limited literature in this area. As this was a scoping review, no formal quality assessment of the articles was carried out. Instead, a scoping review aims to identify the research in the field to provide a descriptive summary [28,50].

## 6. Conclusions

Providing early interventions to 0–36-month-olds in which caregivers carry out the activities promotes the young child’s motor development. The caregivers found the interventions valuable and had a positive attitude towards them. Furthermore, the interventions increase the caregivers’ interest and motivation to promote the young child’s motor development, which is essential in maintaining the behaviour after the end of the interventions. Based on our findings, the following elements can advantageously be included in future interventions where the focus is on promoting the young child’s motor development: supervision and guidance for the child’s caregivers to acquire knowledge and skills about age-appropriate behaviours about their child’s motor development, as well as to practice age-appropriate training to build up skills and concretely acquire ideas for implementation. In addition, knowledge about age-appropriate toys is essential for effective interventions. The interaction between the child and the caregivers is essential; therefore, the daily caregivers must carry out the interventions.

## Figures and Tables

**Figure 1 ijerph-19-11543-f001:**
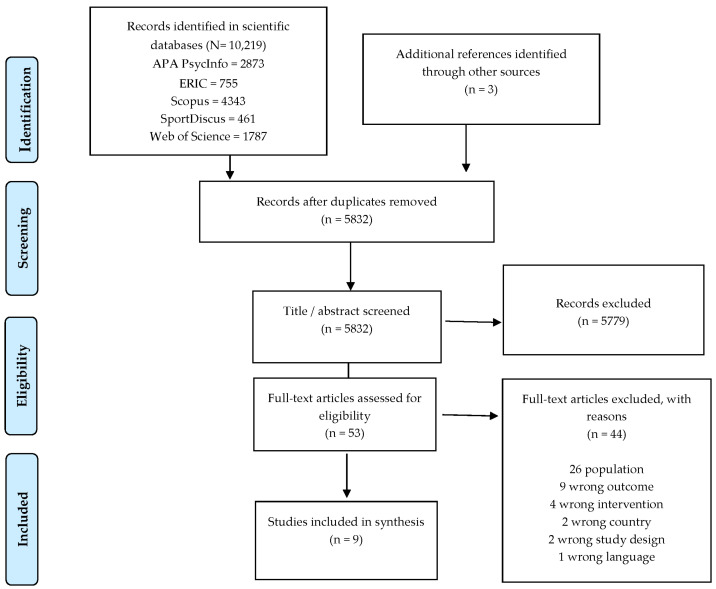
Flow chart of the screening process.

**Table 1 ijerph-19-11543-t001:** Inclusion and exclusion criteria of the study.

Inclusion	Exclusion
**Population**	**Population**
0–36-month-old children The ‘ordinary’ (physically well, ‘healthy’) population of young children	Above 36 months Specific groups of children with specific conditions, e.g., autism, Down’s syndrome, cerebral palsy or similar
**Type of intervention**	**Type of intervention**
Takes place in the child’s ‘natural’ environment with the child’s carers making an effort to promote the child’s motor development. For example, parents at home who participate in the intervention or day care staff in a nursery or similar. The aim is to improve the motor development of the 0–36-month-old child. The content of the intervention itself can be of different types and duration	Not ‘natural’ (e.g., specialised training, such as by physiotherapists)
**Caregivers**	**Caregivers**
Parents Daycare staff	Other caregivers, e.g., physiotherapist, occupational therapist or doctor, are excluded. Defined groups that do not correspond to the general normal population of carers (e.g., only parents with certain conditions in vulnerable areas or with an illness, etc.)
**Geographical and cultural area**	**Geographical and cultural area**
Studies in Western European countries, North America and Australia	Studies from developing/third world countries are excluded as the context and premises are dissimilar, compared to developed countries
**Outcome measure**	**Study type**
Primary outcome measure: impact of interventions on children’s motor development	Everything that is not peer-reviewed scientific articles

**Table 2 ijerph-19-11543-t002:** Search strategy in the five databases.

Search Strategy	
SPORTDiscus	
**Study type**	((real-life or exploratory or natural* or “Community based” or outreach or intervention* OR prospective) N6 (trial* OR experiment* or investigation* or program* or effort or stud*))
	**AND**
**Setting**	[“home session”] OR “home environment*” OR “family environment*” OR “child care” OR daycare* OR “child day care” OR nurser* OR care-house* OR “care provider*” OR care-giver* OR “care giver*” OR “health visitor*” OR parent* OR mother* OR father* or home-based or family-based)
	**AND**
**Population**	(baby OR babies OR child* or toddler* or infant* or “Early childhood” OR newborn*)
	**AND**
**Outcome measure**	(“motor activit*” OR “motor performance” OR “Motor skill*” or “motor development*” or “physical activity” OR movement* OR DE “MOTOR ability in children” OR DE “CHILD development”)
**Scopus**	
**Study type**	((TITLE-ABS-KEY (real-life OR exploratory OR natural* OR “community based” OR outreach OR intervention* OR prospective)) W/6 (TITLE-ABS-KEY (trial* OR experiment* OR investigation* OR program* OR effort OR study*))
	**AND**
**Setting**	(TITLE-ABS-KEY “home session*” OR “home environment*” OR “family environment*” OR “child care” OR daycare* OR “child day care” OR nurser* OR care-house* OR “care provider*” OR care-giver* OR “care giver*” OR “health visitor*” OR parent* OR mother* OR father* or “home-based” or “family-based”))
	**AND**
**Population**	(TITLE-ABS-KEY (baby OR babies OR child* OR toddler* OR infant* OR “Early childhood” OR newborn*)
	**AND**
**Outcome measure**	(TITLE-ABS-KEY (“motor skill*” OR “motor activit*” OR “motor development*” OR “motor performance” OR “physical activity” OR movement*)
**APA Psycinfo**	
**Study type**	((real-life or exploratory or natural* or community based or outreach or intervention* or prospective) adj6 (trial* or experiment* or investigation* or program* or effort* or stud*)).mp.
	**AND**
**Setting**	((home session* or home environment* or family environment*).mp. or home environment/or exp child care/or daycare*.mp. or child day care/or nurser*.mp. or care-house*.mp. or care provider*.mp. or care-giver*.mp. or care giver*.mp. or health visitor*.mp. or parent*.mp. or mother*.mp. or father*.mp. or home-based.mp. or family-based.mp.)
	**AND**
**Population**	(baby or babies or child* or toddler* or infant* or Early childhood or newborn*).mp.
	**AND**
**Outcome measure**	(exp Early Childhood Development/or Motor performance or motor activit* or Motor skill* or motor development* or physical activity or movement*)
**ERIC**	
**Study type**	((“real-life” OR exploratory OR natural* OR “community based” OR outreach OR intervention* OR prospective) N6 (trial* OR experiment* or investigation*or program* OR effort* OR stud*))
	**AND**
**Setting**	(DE “Family Environment” OR DE “Child Care” OR “home session*” OR “Home environment*” OR “family environment*” OR daycare* OR “child day care” OR nurser* OR “care-house*” OR “care provider*” OR “care-giver*” OR “care giver*” OR “health visitor*” OR parent* OR mother* OR father* or home-based or family-based)
	**AND**
**Population**	(baby OR babies OR child* OR toddler* or infant* OR “early childhood” OR newborn*)
	**AND**
**Outcome measure**	(“Motor performance” OR “motor activit*” OR “motor skill*” OR “motor development*” OR “physical activity” OR movement* OR DE “Psychomotor Skills” OR DE “Motor Development”)
**Web of Science**	
**Study type**	((real-life OR exploratory OR natural* OR “community based” OR outreach OR intervention* OR prospective) NEAR/6 (trial* OR experiment* OR investigation* OR program* OR effort OR stud*))
	**AND**
**Setting**	((“family environment*” OR “home session*” OR “home environment*” OR “child* care” OR daycare* OR “child* day care” OR nurser* OR care-house* OR “care provider*” OR care-giver* OR “health visitor*” OR “care giver*” OR parent* OR mother* OR father* or “home-based” or “family-based”))
	**AND**
**Population**	((baby OR babies OR child* OR toddler* OR infant* OR “Early childhood” OR newborn*))
	**AND**
**Outcome measure**	(“Motor skill*” OR “motor activit*” OR “motor development*” OR “motor performance” OR “physical activity” OR movement*)

**Table 3 ijerph-19-11543-t003:** Included studies.

- First Author - Country - Year of Publication	Study Type	- Study Population - Age	- Intervention - Control	Duration	Variables	Primary Outcome Measure: Motor Development Outcome Measure	Secondary Outcome Measure: Carers’ Attitudes to and Their Benefits of the Intervention
[29] - Gross - 2017 - USA	Randomised control trial	- Infants Interven-tion (n = 221) Control (n = 235) - 0 months	- An intervention that focuses on increasing the parents’ skills and knowledge in promoting the child’s motor development, e.g., by having the child on its stomach/tummy time, or having the parents place a toy in front of the child while the child is lying on its stomach. - The ordinary everyday life.	3 months	Motor milestone	In the intervention group, the infants spent significantly more time tolerating the prone position on the floor, compared to the control group.	-
[36] - Hewitt - 2020 - Australia	Randomised control trial	- Infants Interven-tion (n = 16) Control (n = 19) - 0–12 weeks	- Mothers received group tummy time classes with their child and family health nurse for 4 weeks at 2 h of lessons. The mothers have to practice with their child at home; moreover, they get messages about practising with their child three times per week. - Mothers received the usual care from their child and family health nurse.	4 weeks	Motor development was collected at baseline, post-intervention and again when the infants were approximately 6 months old (follow-up).	Post-intervention, there was a moderate effect size for the infant’s ability in prone and sit, favouring the intervention group.	The mothers received the intervention well and found the group practice valuable and relevant.
[31] - Lee - USA - 2012	Non-randomised control trial	- Infants Intervention (n = 11) Control (n = 11) - 1 month	- Caregiver carries out daily activities with a child for 20 min, focusing on improving neck, back, arm and body muscles. Includes rattle and grasp ball. - Caregiver conducts face-to-face communication with the child for 20 min. daily	1 month	Positioning and head control are measured every two weeks until the baby is 4 months old.	The intervention group has better head control and positioning during the training period and after training stops.	-
[30] - Lobo - 2012 - USA	Non-randomised control trial	- Infants Intervention (n = 14) Control (n = 14) - 2 months	- The caregiver carries out a 15-min daily home exercise programme with the child, focusing on back strength and head control (handling and positioning). - The caregiver engages in 15 min of daily face-to-face interaction with the child, with the child lying on its back.	3 weeks	The child’s motor development was measured immediately after the intervention and with subsequent follow-up tests until the child was 15 months old.	The intervention group scored better in the motor milestone measurements both immediately after the intervention ended and in the follow-up measurements.	The families showed positivity and a greater interest in play and the use of toys with the child
[35] - Mendelsohn - 2007 - USA	A single-blind randomised control trial	- Infants to toddlers intervention (n = 52) Control (n = 47) - 0–33 months	- Until the child is 33 months old, the intervention group received video recorded sessions, where a child-development-specialist gives the caregivers feedback on how they react to and what to do with the child. Hereafter, the caregivers get activities to practice at home. They also receive developmentally materials, e.g., toys. The program is based on parenting activities, teaching and playing with their child. - Their normal practice.	33 months	Motor development and parents’ development were measured.	The intervention group was less likely to have motor development delays than the control group.	The caregivers in the intervention group had improved practising with their child and had a stress reduction, compared to the control group.
[32] - Mendres-Smith - 2020 - USA	Observational study with intervention	- Infants (n = 7) - 7 weeks to 4 months	- Two different initiatives: Experiment (1) was guided by the experimenter, the child was laid on its stomach, and the experimenter interacted with the child on a mat (performed at least one day a week for five min duration). Experiment (2) each mother placed the baby breast-to-breast and did different activities with the baby, including a play mat with different features (carried out at least one day a week for five min. duration).	4 months	The child’s head elevation was measured, as well as knowledge about the child’s contentment (e.g., whining or crying).	Experiment (1) no evidence Experiment (2) increase in the child’s motor milestone in head elevation and a reduction in the child’s whining and crying.	The mothers responded that they found Experiment (2) more effective and favourable.
[37] - Palmer - 2019 - USA	Randomised control trial	- Infants and their caregivers Interven-tion (n = 23) Control (n = 19) - 2–5 months of age	- The intervention group get movement lessons about how to guide and help their infants to find new movement possibilities, including positions and use of toys, with a focus on bringing the baby from his or her back to the prone.	One week	Motor-milestone	In the intervention group, the infants spent significantly more time, tolerating the prone position in the week following the lesson.	The caregivers in the intervention group had higher values, indicating more reported knowledge and enjoyment of interacting with the infants.
[34] - Schaub - 2019 - Switzer-land	Randomised control trial	- Infants and toddlers Interven-tion: (n = 109) Control: (n = 102) - 0 to 36 months.	- The caregivers get a training program for their young child at home, which a qualified parents educator carries out. The caregivers get a minimum of 10 visits per year and at least one group meeting per year. The parents get activities/training program each visit and knowledge about early motor childhood development and the improvement of parental practices. -	36 months	Motor-milestones	The intervention group had a more significant proportion of developmental milestones and self-help skills, compared to the control group.	-
[33] - Veldman - 2015 - Australia	Non-randomised control trial	- Toddlers Interven-tion: children from two daycare centres (n = 32) Control: children from two other daycare centres (n = 28) - 24 months	- The caregivers carry out 10 min of daily activities with the children, alternating between elements of exploration (e.g., jumping) and balance and extension activities (e.g., kicking). - Their normal practice	8 weeks	The child’s motor development was measured immediately after the intervention.	The intervention group improved their motor development, compared to the control group.	The caregivers were positive and motivated about the implementation of the intervention.

**Table 4 ijerph-19-11543-t004:** General overview of the results according to interventions addressed to 0–36-year-old children.

	Impact on Motor Development	Impact on the Carers’ Attitudes and Benefits of the Intervention	Duration and Dosage of the Interventions	Factors in the Interventions across the Studies
Interventions addressed to 0–36-year-old children.	Overall, the interventions promote the young child’s motor development	Increases the caregiver’s motivation and interest in the young child’s motor development.	Different across the studies	- Knowledge of behaviour to promote the young child’s motor development - Help, guidance, and supervision to the caregivers - To get practice demonstration in their everyday surroundings - Concrete/inspiration to activities that promote the young child’s motor development according to the child’s age - Toys can be an element in the interventions to implement and promote the young child’s motor development - Preferably including the child’s carer(s) who are responsible for carrying out the intervention with the child

## Data Availability

Not applicable.

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
