# Peer review of "Interventions by Caregivers to Promote Motor Development in Young Children, the Caregivers’ Attitudes and Benefits Hereof: A Scoping Review"

_ijerph, 2022, doi:10.3390/ijerph191811543_

Round 1

Reviewer 1 Report (New Reviewer)

This paper is trying to evaluate the effects of introducing physical activities in the first 36 months of babies. Only 9 studies with three different designs are studied and supported the effects of intervention.

I like to be discussed if the RCT studies demonstrate a more significant effect than the other studies.

Author Response

Reviewer 2 Report (New Reviewer)

You have attempted to perform a scoping review on a relevant topic area. When writing please think of your two aims constantly, the reason I say this is because the caregivers’ experiences, benefits and attitudes towards the interventions seem to be forgotten at times, this second aim especially needs be addressed/addressed more in the discussion and conclusion. 

Author Response

Reviewer 3 Report (New Reviewer)

General comment

This was a revised version of a review aimed at providing an updated state of the art concerning the Interventions by caregivers to promote motor development in very young children aged from 0 to 3 years. The review has been well conducted and carefully revised, and the information herein reported can be useful for all operators working in this field in the aim of more properly improving the approach with children in the early beginning of their lives. In my opinion, the manuscript in its present form can be considered for publication in this journal as it is, I only have some minor indications just related to text editing, as reported below.

Minor comments

-Table 3, reference to paper n 36.

…Mothers received group

-Table 3, reference to paper n 34, second column

Rrandomised contro trial…..To be fixed: change to Randomised

 -Table 3, reference to paper n 34, last but one column

The intervention group (it was: interventiongroup)

Compared to the control group (it was: compared to the controlgroup)

Author Response

This manuscript is a resubmission of an earlier submission. The following is a list of the peer review reports and author responses from that submission.

Round 1

Reviewer 1 Report

I would recommend extend the conclusions because they have a lot of information about the topic. The results are very descriptive.

Reviewer 2 Report

The manuscript summarizes the scientific evidence that exists in some databases on the influence of caregivers on children's motor development between 0 and 2 years old. This review could be considered relevant for this journal if it were more relevant databases, such as Web Of Science, and if there were a large number of articles that needed this type of review, because there are only five studies and it is insufficient. For this reason, it is considered that this manuscript does not meet the quality criteria of the journal (in reference to relevance and scientific interest).

To improve this manuscript, here are some suggestions:

General

Throughout the manuscript it is mentioned that this review is for children from 0 to 2 years old, but there are only 4 manuscripts that study infants 7 months or less and 1 manuscript with infants 2 years old. There is a lot of temporal difference between one and the other. In addition, as is known, it is the time with the greatest changes in the development of the person. Therefore, generalizing infants from 0 to 2 years old with only these 5 manuscripts may not be realistic.

Introduction

The "state of the art" of the manuscript should be improved, because no problem is explained that justifies the reason for performing this review. The authors mention that the reason for carrying out this review is because it has not been done before, but this is not an adequate justification. In fact, it probably hasn't been done before because the number of studies on this topic is really small.

Methods

- If we would like to publish with the highest quality, we must include the most relevant databases in the review. In this case, Web Of Science should be included.

- Justify the filters applied in the review. For example, why have manuscripts between 2000 and 2022 been chosen? Why have these languages been chosen?

Results

Wrong outcome, wrong intervention, wrong country, wrong study design and wrong language must be explained.

Discussion

- 2 paragraph: It is recommended to rewrite the second paragraph because it is an idea reiterated in the introduction.

- Lines 73-75: This sentence should be rewritten because we cannot conclude that the results of the review are reliable as other studies with other samples have found similar results. It is an overrated statement.

Reviewer 3 Report

Thank you for the opportunity to review the manuscript entitled, "Interventions by Caregivers to Promote Motor Skills in Young Children: A Scoping Review”.

I believe this study investigated a topic relevant to the readers of “IJERPH”.  The purpose of this scoping review is to provide an account of what studies have been published to date about the impact of different types of interventions to promote the motor skills of 0-2-year-olds carried out by the child’s caregivers. Systematic reviews (SRs) and meta-analyses (MAs) are considered as the best tools to synthesize the scientific evidence as to which treatments, interventions, or prevention programs should be applied for a given problem.

Finally, the manuscript does not meet my expectations for publication in the journal.

The study not enough of an advance or of enough impact for the investigation about the promotion Motor Skills in Young Children.

Only studies in Western European countries, North America and Australia are included.

The results of this review are based on only five articles.

In four of the articles, the results are based on relatively few respondents.

The secondary aim is to find out what the carers’ opinion is of the action taken, but only three of the studies contained information about caregivers’ attitudes to and experiences of the intervention.

The search of the scientific literature should be expanded to include others databases:  Web of Science, Science Direct… I thing that a “Scoping Review” with very few included studies is less informative.

The systematic reviews should provide answers to: What intervention is most effective?,  What duration is most effective?...

Thank you. 

Round 2

Reviewer 2 Report

It has been a pleasure reviewing this study again. The authors have made a good effort and addressed most of the recommendations provided. However, the manuscript does not meet the quality expectations to be published in this journal, as already mentioned in the previous review. Therefore, it is considered that the study does not have enough impact to be published in the journal.

I recommend that you submit your manuscript to a lower impact journal.

Reviewer 3 Report

Thank you for the opportunity to review again the manuscript entitled, "Interventions by Caregivers to Promote Motor Development in Young Children, the Caregivers’ Attitudes and Benefits hereof: A Scoping Review”

The authors took into account the comments and they proceeded with the revisions. I thank the authors their time and effort.

Finally, the manuscript does not meet my expectations for publication in the journal. The study not enough of an advance or of enough impact for the investigation about the promotion Motor Development in Young Children.

Thank you.